# Organelle Imaging with Terahertz Scattering-Type Scanning Near-Field Microscope

**DOI:** 10.3390/ijms241713630

**Published:** 2023-09-04

**Authors:** Jie Huang, Jie Wang, Linghui Guo, Dianxing Wu, Shihan Yan, Tianying Chang, Hongliang Cui

**Affiliations:** 1State Key Laboratory of Rice Biology, Institute of Nuclear Agricultural Sciences, Zhejiang University, Hangzhou 310029, China; 12016005@zju.edu.cn (J.H.); dxwu@zju.edu.cn (D.W.); 2Chongqing Institute of Green and Intelligent Technology, Chinese Academy of Sciences, Chongqing 400714, China; wangjie@zucc.edu.cn (J.W.); hcui@jlu.edu.cn (H.C.); 3National Key Laboratory for Germplasm Innovation & Utilization of Horticultural Crops, Wuhan 430070, China; glh@webmail.hzau.edu.cn; 4Shenzhen Institute of Advanced Technology, Chinese Academy of Sciences, Shenzhen 518055, China

**Keywords:** chloroplast, THz imaging, s-SNOM, *Arabidopsis thaliana*, *Camellia sinensis*, flavonoid

## Abstract

Organelles play core roles in living beings, especially in internal cellular actions, but the hidden information inside the cell is difficult to extract in a label-free manner. In recent years, terahertz (THz) imaging has attracted much attention because of its penetration depth in nonpolar and non-metallic materials and label-free, non-invasive and non-ionizing ability to obtain the interior information of bio-samples. However, the low spatial resolution of traditional far-field THz imaging systems and the weak dielectric contrast of biological samples hinder the application of this technology in the biological field. In this paper, we used an advanced THz scattering near-field imaging method for detecting chloroplasts on gold substrate with nano-flatness combined with an image processing method to remove the background noise and successfully obtained the subcellular-grade internal reticular structure from an Arabidopsis chloroplast THz image. In contrast, little inner information could be observed in the tea chloroplast in similar THz images. Further, transmission electron microscopy (TEM) and mass spectroscopy (MS) were also used to detect structural and chemical differences inside the chloroplasts of Arabidopsis and tea plants. The preliminary results suggested that the interspecific different THz information is related to the internal spatial structures of chloroplasts and metabolite differences among species. Therefore, this method could open a new way to study the structure of individual organelles.

## 1. Introductions

Organelles are crucial components in biological cells, exercising core functions in the entire cellular life cycle. The detection and analysis of internal morphological and constituent parameters reveal basic information about life activities and are also an important basis for disease diagnosis [1]. In recent decades, subcellular organelle imaging has provided a new perspective for observing the microscopic world and studying life phenomena inside the cell. However, in optical biological imaging, due to the limited penetration depth of visible light, it can only detect the internal information in organelle samples destructively. In addition, due to the lack of chemical sensitivity of visible light, the physical and chemical information of the target can be identified only with the help of special fluorescent probes. However, the contrast agents used in fluorescence technology can seriously affect the morphology, metabolism and movement of organisms and may lead to cytotoxicity and phototoxicity [2]. These limitations and concerns have stimulated interest in actively developing label-free optical imaging alternatives.

Terahertz radiation refers to the frequency band of 0.1~10 THz in the electromagnetic spectrum, corresponding to the wavelength of 3 mm~30 μm. Compared with visible light, THz can penetrate deeper into bio-matter reaching 100 microns for biological samples [3]. In addition, biological molecules such as proteins, nucleic acids and sugars exhibit unique vibration, rotation and conformation responses to THz radiation, as shown in ref. [4]. Therefore, abundant physical and chemical information in biological samples can be provided in a non-tagged, non-invasive and non-ionizing manner [5,6]. Particularly, the low photon energy of THz radiation causes little ionization damage to biological samples and is a safe and benign detection method at not too high a dose [7,8]. Recently, THz radiation has been widely applied in tissue imaging and achieved great progress in the diagnosis of skin tissues [9,10], corneal diseases [11,12,13,14] and tumor boundaries [15,16,17].

As a diagnostic indicator, a THz image provides a faithful and vivid description or portrait of a biological object, with its information capacity relating directly to its reliability in discerning minute dielectric differences. Generally, the higher the resolution, the more details of the original scene the image can provide. However, traditional THz far-field imaging methods are bound by the Rayleigh diffraction criterion, with a limited resolution of millimeters to sub-millimeters. In the face of the increasing demand for diagnostic accuracy in precision medicine and early diagnosis, it is urgent and significant to break through the diffraction limit and develop THz super-resolution imaging methods. Accompanied by the development and application of near-field scattering and imaging theory in the THz band, a series of THz super-resolution imaging methods have been developed whose resolution reaches the micrometric level or beyond. For all that, they still need a variety of experimental verifications before practical application.

At the cellular level, the dehydration state of a single plant cell (~100 μm) has been monitored and characterized by terahertz near-field imaging technology based on the microstructure photo-conductive antenna [18]. In 2022, a single onion epidermal cell in the onion epidermis was imaged in vivo at an amplitude signal of around 1 THz [19]. Apart from being exploited to reveal physiological states, biological activities and cellular heterogeneities of various types of cells, differentiated THz dielectric response shows clearly the distinctive intracellular and intercellular regions, revealing great latent capacity in subcellular structure identification [18,19]. Nevertheless, since the resolution for this scheme based on PCAM depends heavily on the size of the antenna and distance control, it is difficult to continue increasing the resolution [20].

On the other hand, the developing THz scattering-type scanning near-field optical microscope (THz s-SNOM) has the highest resolution at nanometer level in the field of THz imaging, showing great potential to achieve a fine description of organelles without damage [21]. The THz s-SNOM is an innovative, ultra-high-resolution optical microscope and spectral system based on atomic force microscopy (AFM) technology [22]. The THz s-SNOM system relies on light scattering from metallized probes of nanoscale AFM to achieve near-field optical imaging and spectroscopy at subwavelength-scale spatial resolution, independent of the wavelength of light. The s-SNOM, as a powerful tool for optical imaging, has nanometer resolution but with a relatively low signal-to-noise ratio (SNR) and narrow bandwidth [23]. With the discovery of graphene-based sample substrates with extremely high flatness and sufficient THz reflectivity, single protein molecules with nanometer size have been imaged and analyzed [21]. The success of this experiment makes it possible to study single organelles using the THz s-SNOM system.

Chloroplast is a specific organelle with autofluorescence which exists in higher plants and some bacteria. As the main organelle of photosynthesis, chloroplast contains various chemicals and structures inside [24]. Optical and fluorescent microscopes are usually used for nondestructive chloroplast observation but lack inner information about the chloroplast. Scanning and transmission electron microscopes can provide inner structures from a slice of chloroplasts [25,26]. The present work is aimed at developing a new method for studying single Arabidopsis chloroplasts using the THz s-SNOM. Through the coupling effect of the nanoscale radius of a platinum probe and gold substrate, we could obtain a greatly enhanced terahertz near-field signal from a biological sample; combining this with an image processing method to remove the background noise and performing segmentation to obtain the subject target, we could further obtain the micron-grade internal structure of the chloroplast terahertz spectral image. However, little inner information about the tea chloroplast was observed using the same THz s-SNOM. Further transmission electron microscope (TEM) analyses of leaves from *Arabidopsis thaliana* and *Camellia sinensis* revealed little ultrastructural differences, but metabolome analyses showed great metabolic differences in flavonoids among species. The present research opens a new way to study the structure of individual organelles.

## 2. Results

### 2.1. Chloroplast Imaging of Arabidopsis thaliana and Camellia sinensis with Fluorescent Microscope

The strategy for organelle imaging with THz is shown in Figure 1. As the most commonly used model plant in biological research, *Arabidopsis thaliana* is used for chloroplast imaging with fluorescent microscope [27]. After Percoll continuous density gradient, Arabidopsis chloroplasts were separated into a band (Figure 2a). Chloroplast images in optical and fluorescent fields were immediately analyzed with a fluorescent microscope. At 400× or 630× magnification, a large number of Arabidopsis chloroplasts with red autofluorescence were observed, indicating the effectiveness and feasibility of the isolation process (Figure 2b). However, little details in Arabidopsis chloroplasts could be seen in the optical and fluorescent fields. To evaluate the effect of formaldehyde fixation on chloroplast morphology, images of fixed chloroplasts were obtained in the optical and fluorescent fields. Arabidopsis chloroplasts after formaldehyde fixation showed no differences to the unfixed chloroplasts (Figure 2b). Thus, the formaldehyde fixation process was used for further experiments which were more stable in structure. In addition, we could see from the figure that the isolated chloroplast had high purity, which ensured the accuracy of the THz spectrum detection of the target species and avoided the interference of cell fragments with comparable size.

*Camellia sinensis* is an evergreen and perennial species containing various metabolites in its leaves [28]. To investigate the influence of its abundant metabolites, two representative cultivars were chosen for the THz imaging, ‘*Echa 10*’ and ‘*Shuchazao*’. The centrifugal force of 8000 *g* was more suitable for the chloroplast isolation of *Camellia sinensis* since 18,000 *g* caused obvious chloroplast damage and polyphenol leakage (Figure 3a). Compared with chloroplasts of *Arabidopsis thaliana*, chloroplasts of *Camellia sinensis* were surrounded by a red substance during the isolation process, which was probably the oxidation product of tea polyphenols (Figure 2a and Figure 3a). Nevertheless, the purity of chloroplast separation was very high. Chloroplast images of two tea cultivars were analyzed in optical and fluorescent fields with the fluorescent microscope. Similar to the results of Arabidopsis chloroplasts, tea chloroplasts with red autofluorescence could be clearly observed with little details in the optical and fluorescent fields (Figure 3b).

### 2.2. Chloroplast Imaging of Arabidopsis thaliana and Camellia sinensis with the THz s-SNOM

The THz s-SNOM uses a combination of scattering THz spectroscopy and AFM. Figure 4 shows the results of AFM imaging of a single chloroplast, where Figure 4a is a topographic map which clearly shows the morphological information of a single chloroplast. The single chloroplast tested here was approximately round, and it became higher from the edge to the middle. Figure 4b is the amplitude image, which clearly shows the uneven surface of the chloroplast. The profile line (Figure 4c) shows that the height and diameter of chloroplast were about 0.7 μm and 5.5 μm. Figure 5 shows the THz near-field amplitude and phase imaging at different harmonic demodulations (from 2 Ω to 5 Ω). Using the topographic map of AFM as a standard reference (Figure 4a), we observed that the THz near-field amplitude and phase imaging were consistent, and both could accurately describe the shape and size of individual chloroplasts. As can be seen in the test results shown in Figure 5, in general, a clear regional division between the biological sample and the substrate could be easily identified based on the THz near-field images. On the other hand, as the order increased, the demodulated phase image became clearer. Compared with atomic images, THz spectral images can not only describe the morphology of chloroplasts, but, more importantly, they can look into the chloroplasts and describe the distribution of substances inside chloroplasts according to the differences in the composition and dielectric properties of the chloroplasts. Figure 6 shows the THz near-field images after denoising and segmentation. The THz image after demodulation of the higher-order signal could clearly describe the distribution of material inside the chloroplast.

Afterwards, chloroplasts of two tea cultivars were used for THz imaging. In the atomic force field, height and amplitude images of chloroplasts from two cultivars were obtained (Figure 7). The chloroplast diameters of tea cultivars were similar to those of the Arabidopsis chloroplast, with heights of nearly 0.5 μm (Figure 7a,b). The chloroplast surfaces of the two cultivars were clearly observed in the AFM amplitude images (Figure 7c,d). In the THz field, amplitude and phase images of the tea chloroplast were obtained. Similar to the THz amplitude image results for Arabidopsis chloroplasts, chloroplasts of two tea cultivars could be distinguished from the background, and little details were seen inside the chloroplasts in 2 Ω to 5 Ω (Figure 8a–d). However, the chloroplasts of the two tea cultivars could not be seen in the 2 Ω to 5 Ω THz phase images, suggesting the presence of a specific interference source during the THz phase imaging process (Figure 8e–h).

### 2.3. Ultrastructure Imaging of Leaves from Arabidopsis thaliana and Camellia sinensis

Since significant differences existed between the chloroplast THz imaging results for *Arabidopsis thaliana* and *Camellia sinensis*, we performed ultrastructure imaging of leaf slices with TEM to evaluate their morphological diversities. TEM results revealed abundant internal structures in chloroplasts, including thylakoids (Figure 9). Chloroplasts were an ellipse shape and slightly varied according to their position in a cell. The major axis of the chloroplasts was from 4 μm to 7 μm, and the minor axis was nearly 2 μm. There were little differences in size between chloroplasts from the two species.

However, at the same time, the internal morphological characteristics were different between the two species. It can be seen from the TEM pictures that the thylakoids in the chloroplasts of Arabidopsis thaliana gathered in bands along the long axis of the chloroplasts, and were distributed evenly and loosely in the stroma, and there was a certain gap between the bands of different types of thylakoids. In contrast, the density of thylakoids in the chloroplast of the tea plant was higher, and the number of stroma was smaller. There was no obvious aggregation, the layer was filamentous, the arrangement direction was not parallel to the long axis and the center of a point in the interior was radially divergent. There were obvious differences in the distribution patterns of thylakoids and stroma in the chloroplast from these two different plants (Figure 9).

### 2.4. Metabolome Analyses of Leaves from Arabidopsis thaliana and Camellia sinensis

Further metabolome analyses provided a different perspective on the material composition difference of the two species. Three different samples were individually accumulated in the PCA analysis of the metabolome, indicating the data reliability (Figure 10a). In addition, PC1 was the major factor causing the difference between *Arabidopsis thaliana* and *Camellia sinensis*. A total of 938 metabolites were identified in three samples via metabolome analyses (Figure 10b,c). In ‘*Echa 10*’ vs. *Arabidopsis thaliana*, 443 and 216 metabolites were respectively up-regulated and down-regulated (Figure 10b and Appendix A). In ‘*Shuchazao*’ vs. *Arabidopsis thaliana*, there were 403 up-regulated and 257 down-regulated metabolites (Figure 10c and Appendix A). With the screening criterion of fold change > 1.5 or *p*-value < 0.05, 78 and 96 differential metabolites were individually picked out in ‘*Echa 10*’ vs. *Arabidopsis thaliana* and ‘*Shuchazao*’ vs. *Arabidopsis thaliana* (Appendix A). Since the THz imaging with chloroplasts from *Camellia sinensis* was unsuccessful, commonly up-regulated differential metabolites in ‘*Echa 10*’ and ‘*Shuchazao*’ were focused on. A total of 38 commonly up-regulated differential metabolites are listed in Appendix A, which were probably related to the THz imaging of the tea chloroplasts (Appendix A). Classifications of differential metabolites revealed three common groups in ‘*Echa 10*’ vs. *Arabidopsis thaliana* and ‘*Shuchazao*’ vs. *Arabidopsis thaliana*, ‘flavonoids’, ‘carboxylic acids and derivatives’ and ‘organooxygen compounds’ (Figure 10d,e). Additionally, two groups of metabolites specifically existed in ‘*Echa 10*’ vs. *Arabidopsis thaliana*, ‘benzene and substituted derivatives’ (6.49%) and ‘fatty acyls’ (5.19%). Differential metabolites were further MetPA analyzed to screen key metabolites affecting chloroplast THz imaging in *Camellia sinensis*. The results showed that metabolites related to ‘flavonoid biosynthesis’, ‘flavone and flavanol biosynthesis’ and ‘propanoate metabolism’ pathways were significantly accumulated in ‘*Echa 10*’ vs. *Arabidopsis thaliana* and ‘*Shuchazao*’ vs. *Arabidopsis thaliana* (Figure 11). In these three pathways, five commonly up-regulated metabolites were found, dihydrokaempferol, myricetin, quercetin, (s)-methylmalonic acid semialdehyde and succinic acid (Appendix A).

## 3. Discussion

As crucial components in biological cells, organelles exercise core functions in the entire cellular life cycle. In recent decades, subcellular organelle imaging has provided a new perspective for studying life phenomena inside the cell. However, optical biological imaging can only detect the internal information in organelle samples destructively due to the limited penetration depth of visible light [27]. In this study, little details in Arabidopsis chloroplasts could be seen in optical and fluorescent fields (Figure 2 and Figure 3), which is similar to the previously reported results [27,29]. Compared with visible light, THz can penetrate deeper into bio-matter and can reach 100 microns for biological samples [3]. In addition, biological molecules such as proteins, nucleic acids and sugars exhibit unique vibration, rotation and conformation responses to THz radiation [4]. Therefore, abundant physical and chemical information in biological samples can be provided in a non-tagged, non-invasive and non-ionizing manner [5,6]. In the present study, THz imaging of Arabidopsis chloroplasts with the THz s-SNOM displayed abundant inner information in a non-tagged manner (Figure 4, Figure 5 and Figure 6), which is a great breakthrough in the organelle imaging field. The THz phase information supplies the relative optical depth profile of the sample’s surface and interior density [30]. The chloroplast is usually a reticular structure connected by the placode thylakoids between the granulated thylakoids [31]. The grana are about 0.25 to 0.8 um in diameter and consist of 10 to 100 thylakoids. There are about 40 to 60 grana in each chloroplast. The image-processed THz near-field signal presented here could clearly delineate the internal network of chloroplasts at a level of detail similar to the size of a granum, indicating the great possibility of THz imaging for inner information acquirement in plant chloroplasts in a label-free, non-invasive and non-ionizing manner.

*Arabidopsis thaliana* is an annual model plant commonly used for scientific research [27], and *Camellia sinensis* is an evergreen and perennial species containing various metabolites in the leaves [28]. In this study, the size and external morphology of the chloroplasts were similar between the two species in the optical and fluorescent fields (Figure 2 and Figure 3) and corresponded to typical chloroplasts [27]. For biological samples, internal spatial structure and specific metabolites can be detected by THz phase images [32]. The THz image after demodulation of the higher-order signal could clearly describe the distribution of material inside the Arabidopsis chloroplast (Figure 4, Figure 5 and Figure 6) but not that of the tea chloroplast (Figure 7 and Figure 8). Each lamella of the thylakoid is composed of two closed membranes, the main components of which are lipids, while the stroma is the water liquid between the intima and the thylakoid. Their dielectric characteristics differ greatly in the THz band. Considering the composition and structure information comprehensively, there were obvious differences between the internal paradigm of the chloroplasts of the two plants (Figure 9, Figure 10 and Figure 11). The degree of separation in the chloroplast of Arabidopsis thaliana was higher, while that in the chloroplast of tea seemed to be more uniform. This difference still existed even after fixing. Generally, the greater the structural and chemical information difference, the more easily it is reflected by the imaging results. The difference in THz imaging performance seems to be related to the composition and structure of chloroplasts of different species. Thus, we speculate that the difference between the internal spatial structure and metabolites in chloroplasts of different species leads to different results in the THz imaging of chloroplasts.

## 4. Materials and Methods

### 4.1. Experimental Design

For the assessment of chloroplast images by Pt nanoprobe THz s-SNOM, chloroplast images from fluorescent microscope were used as contrast. As shown in Figure 1a, the THz s-SNOM offers chloroplast images in atomic force and THz fields, while a fluorescent microscope provides those in the optical and fluorescent fields (Figure 1a).

### 4.2. Plant Materials

*Arabidopsis thaliana* is a vital model plant for scientific research, perfect for exploring the possibility of THz imaging of a plant chloroplast in this research [33]. Meanwhile, *Camellia sinensis* is a perennial species characterized by rich secondary metabolites in the leaves, particularly suitable for analyzing the influence of metabolites on chloroplast THz imaging [34]. After being sterilized with 8% NaClO solution, seeds of wild-type *Arabidopsis thaliana* were sowed on base material and cultured for 3 weeks. The cultivation condition was 300 µmol m^−2^ s^−1^ photon flux density for 12 h per day at 20 °C. Arabidopsis leaves were harvested for chloroplast isolation, TEM imaging and metabolome analysis. Two cultivars of *Camellia sinensis* (‘*Shuchazao*’ and ‘*Echa 10*’) were chosen for chloroplast isolation, TEM imaging and metabolome analyses. ‘*Shuchazao*’ is a representative tea cultivar whose genome has been sequenced and assembled [28]. ‘*Echa 10*’ is a representative tea cultivar widely used in scientific research [35]. One bud and two leaves of *Camellia sinensis* were harvested from the germplasm resources nursery of Huazhong Agricultural University, Wuhan, China (30°280′0″ N, 114°220′8″ E). Each sample was independently collected three times from nine randomly selected plants.

### 4.3. Isolation of Plant Chloroplasts

Chloroplasts were isolated from leaves of *Arabidopsis thaliana* or *Camellia sinensis* using Percoll continuous density gradient according to Ling and Jarvis (2016), and all steps were completed on ice [29]. Briefly, 5 mg of reduced glutathiones was added into the mixed liquid of 13 mL Percoll and 13 mL 2 × chloroplast isolation buffer, which was centrifugated with a force of 43,000 *g* at 4 °C for 30 min to prepare the continuous density gradient. A 0.3 g amount of leaves of *Arabidopsis thaliana* or *Camellia sinensis* was ground into homogenate in 1 × chloroplast isolation buffer, and the homogenate was filtered with a double-layer magic filter cloth. The filtrate was centrifugated with a force of 1000 *g* at 4 °C for 5 min, and the precipitation was suspended in the remaining supernatant. The suspended homogenate was transferred to the top of the continuous density gradient and centrifugated with a force of 3000 *g* (*Arabidopsis thaliana*) or 8000 *g* (*Camellia sinensis*) at 4 °C for 10 min. Finally, the band contained complete chloroplasts.

### 4.4. Fluorescent Microscope Imaging

A fluorescent microscope is able to form images via fluorescent substances in biological samples, such as biotin, fluorescein isothiocyanate (FITC) and so on [36]. Chloroplast is a key organelle responsible for photosynthesis in a cell with autofluorescence [31]. Isolated chloroplasts of *Arabidopsis thaliana* or *Camellia sinensis* were individually observed using an inverted fluorescent microscope (Leica DMi8, Leica Microsystems, Germany). Briefly, isolated chloroplasts were resuspended with 1× chloroplast isolation buffer and spread on a slide. Optical images were obtained in 400× and 630× fields, while fluorescent images were captured using the red fluorescent field (excitation wavelength 640 nm, emission wavelength 680 nm). They were used to prove the purity of further THz imaging objects.

### 4.5. THz s-SNOM Setup

The THz s-SNOM uses a combination of scattering THz spectroscopy and atomic force microscopy. The setup of the THz s-SNOM system produced by Neaspec is shown schematically in Figure 1b. Ultrafast pulses produced by the femtosecond laser with a center wavelength of 1.550 μm are incident onto the photoconductive antenna (PCA) to excite the THz pulses. The input THz pulse (*E_i_(t)*) is directed through a beam splitter and a reflector and then focused onto the Pt probe by a special parabolic mirror that can support dual-beam operation. In order to maximize the focus of the invisible THz light on the tip of the needle, a set of guide light is introduced for correction. In this way, the local THz electric field can be significantly enhanced at the tip apex due to the lighting-rod effect and the dipole electric field enhancement effect [23]. In order to enhance the THz near-field signal compared to the background signal, the probe is controlled by atomic force microscope (AFM) to interact with the sample in intermittent contact mode. The scattering THz near-field signal (*E_s_(nΩt)*) is modulated by the probe oscillating at *Ω* frequency and collected by the parabolic mirror. Further, it is collimated and transmitted to the PCA detector by the interference detection scheme using a Michelson interferometer. The THz near-field amplitude and phase signals corresponding to the dielectric properties of the investigated sample can be acquired by using a lock-in amplifier at *nΩ* (*n* = 1, 2, 3…) for signal demodulation. In the process of imaging, the sample is scanned by the probe controlled by the AFM, and the near-field signals of each spatial point are obtained, which can be used for THz image reconstruction. In addition, the surface topography of the sample can be obtained simultaneously.

Due to the weak dielectric response of biological samples, the generated THz scattering near-field signals are weak. We carried out a series of simulations to explore the enhancement effect of the dielectric effect of the substrate on the electric field at the tip [21]. The simulation results showed that, for materials with strong dielectric response, such as metals, the electric field at the tip has a very large magnification, while, when the substrate is a semiconductor or other non-metallic material with insignificant dielectric response, the electric field enhancement is greatly reduced. The simulation confirmed our intuition that one can take advantage of the obvious dielectric response properties between metal and non-metal to improve the contrast of the samples, and, in this way, one can put the non-metal object, e.g., biological test samples of micro-nano features on a metallic substrate, to accentuate the contrast between the two. This can be used to improve the sample recognition and image quality and has been quite effective in the near-field testing of other bands. In this work, we chose an Au substrate for the THz imaging of chloroplast. In addition, the related parameters of the probe, such as material, length and radius of curvature, also affect the strength of the THz scattering near-field signal. Based on the previous simulation and experimental results, a platinum probe with an ultra-long axis (~300 μm) and tip curvature radius of about 50 nm, controlled by an AFM system with a vertical oscillation frequency of 263 kHz, was selected. In addition, the AFM senses the force between the tip probe and sample with atomic-level resolution [37].

### 4.6. THz Image Process

As the THz near-field image is prone to a multitude of noise, artifacts or inappropriate acquisition conditions, it is difficult to accurately obtain clear images of target samples based on raw image data. Digital restoration methods are employed to process the original THz near-field amplitude and phase image. In this vein, a dual-kernel Gaussian filter was proposed to denoise the original image. Considering the two-dimensional information of the image pixel range and spatial location, the dual-core Gaussian filter is designed to effectively improve the image signal-to-noise ratio (SNR) and solve the problem of image blur. The basic idea is to measure the similarity in pixel range and spatial space at the same time in the process of image denoising. Pixels with close spatial distance and a small difference in pixel value are given largser weight, while pixels with large spatial distance and significant difference in pixel value are given smaller weight. Its intuitive effect is reflected in the low contribution degree of pixels with longer spatial distance and larger pixel difference, which is conducive to the suppression of noise interference. The pixels with close spatial distance and small difference in pixel value contribute more, which is conducive to preserving image details, ultimately improving image SNR, reducing blur effect and improving image quality. Further, the GmbCut algorithm is used to segment the image on the foreground/background area of the pixel map.

### 4.7. TEM Imaging

The specimen was fixed with 2.5% glutaraldehyde in phosphate buffer (0.1 M, pH 7.0) for 1 d and 1% osmium tetroxide in phosphate buffer (0.1 M, pH 7.0) for 2 h. The fixed specimen was dehydrated by a graded series of ethanol solutions (30%, 50%, 70%, 80%, 90%, 95% and 100%) for 15 min in each solution and then transferred to absolute acetone for 20 min. The specimen was infiltrated in order with 50% Spurr resin in acetone for 1 h, 75% Spurr resin in acetone for 3 h and 100% Spurr resin overnight. The infiltrated specimen was embedded in Spurr resin and heated at 70 °C for 12 h. The specimen was sectioned with 80 nm thickness in the Leica EM UC7 ultratome, and sections were stained by alkaline lead citrate and uranyl acetate for 10 min, respectively. The prepared sample was observed in 16,000× and 64,000× fields using the Hitachi model H-7650 TEM.

### 4.8. Metabolome Analysis

Samples were individually ground in liquid nitrogen and ultrasonicated in the water, acetonitrile and isopropanol mixture (V/V/V = 1:1:1) at 4 °C for 30 min. The supernatant was lyophilized and redissolved in 200 μL water and acetonitrile mixture (V/V = 1:1). The UHPLC-Q Exactive HFX spectrometer (Thermo, Waltham, MA, USA) was used to detect metabolites in samples. Chromatographic column: Waters HSS T3 (100 × 2.1 mm, 1.8 μm); mobile phase: phase A is 0.1% formic acid water solution, and phase B is 0.1% formic acid acetonitrile solution; flow rate: 0.3 mL/min; column temperature: 40 °C; injection volume: 2 uL; elution gradient: 0–1.0 min A/B (100:0 V/V), 1.0–9.0 min A/B (5:95 V/V), 9–13 min A/B (5:95 V/V), 13.1–17.0 min A/B (100:0 V/V). QC samples were inserted into the sample queue to monitor and evaluate the stability of the system and the reliability of experimental data. The conditions of electric spray ion source (ESI) were as follows: sheath gas: 40 arb; auxiliary gas: 10 arb; ion spray voltage: 3000 V/−2800 V; temperature: 350 °C; temperature of the ion transfer tube: 320 °C. By matching with the retention time, molecular weight (molecular weight error is less than 10 ppm) and secondary fragmentation spectrum in local databases and commercial databases, metabolites in biological samples were qualitatively and quantitatively analyzed. All samples were analyzed using PCA and PLS-DA. Fold change > 1.5 and *p*-value < 0.05 were used for the identification of differential metabolites. Differential metabolites were further classified and MetPA analyzed.

## 5. Conclusions

We have verified that an advanced THz super-resolution imaging method using a scattering near-field microscopic system with a gold substrate of nano-flatness combined with an image processing method to remove the background noise was successfully employed for single Arabidopsis chloroplast imaging. The subcellular internal reticular structure of the observed Arabidopsis chloroplast was detected in the THz image. At the same time, TEM imaging technology found different patterns of material composition distribution in the chloroplasts of two species, revealing that THz imaging can obtain internal information about chloroplasts. Therefore, this method could open a new way to study the structure of individual organelles.

## Figures and Tables

**Figure 1 ijms-24-13630-f001:**
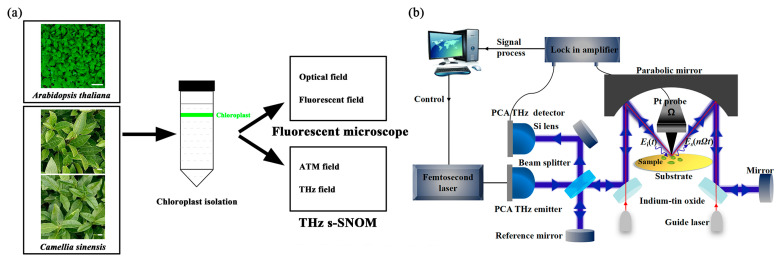
Schematic diagrams of chloroplast isolation (**a**) and chloroplast imaging with the THz s-SNOM (**b**).

**Figure 2 ijms-24-13630-f002:**
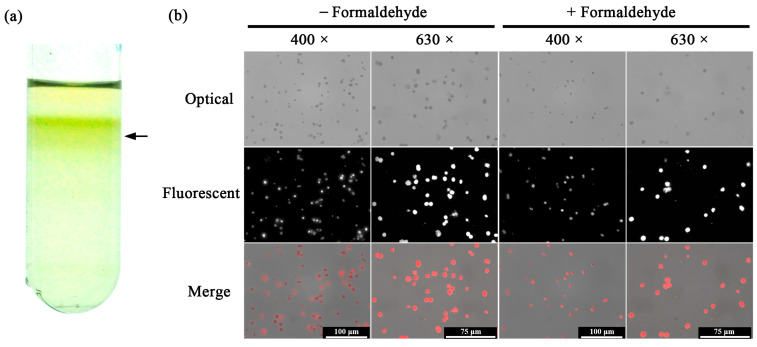
Arabidopsis chloroplasts after application of continuous density gradient (**a**). Optical and fluorescent images of chloroplasts with or without formaldehyde (**b**). Black arrow points to the chloroplasts.

**Figure 3 ijms-24-13630-f003:**
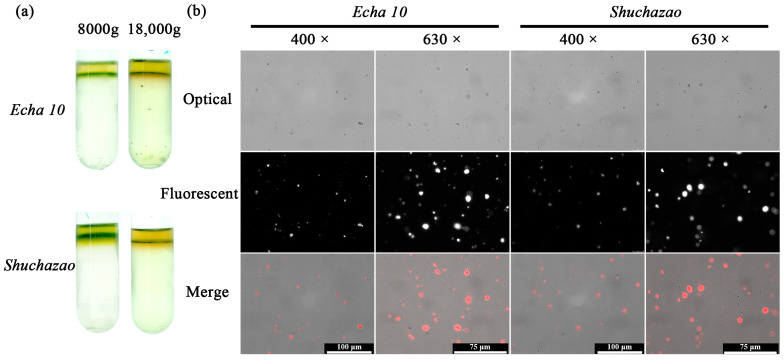
Chloroplasts of *Camellia sinensis* after application of continuous density gradient (**a**). Optical and fluorescent chloroplast images of two tea cultivars (**b**).

**Figure 4 ijms-24-13630-f004:**
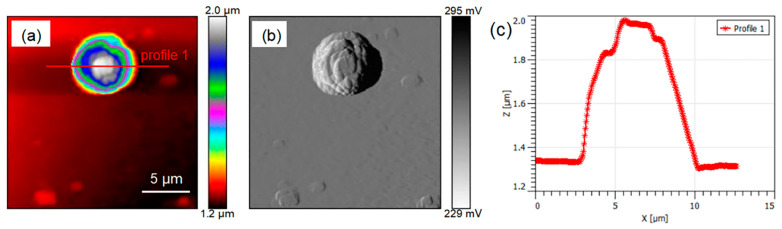
The AFM imaging of individual Arabidopsis chloroplasts. (**a**) Topographical image. (**b**) Amplitude image. (**c**) The section profile along the line indicated in (**a**).

**Figure 5 ijms-24-13630-f005:**
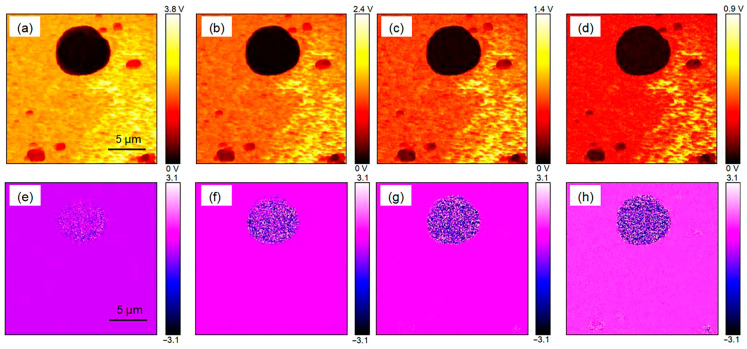
The THz s-SNOM image of individual Arabidopsis chloroplasts. (**a**–**d**) present the near-field amplitude images at 2 Ω, 3 Ω, 4 Ω, 5 Ω; (**e**–**h**) are the near-field phase images at 2 Ω, 3 Ω, 4 Ω, 5 Ω.

**Figure 6 ijms-24-13630-f006:**
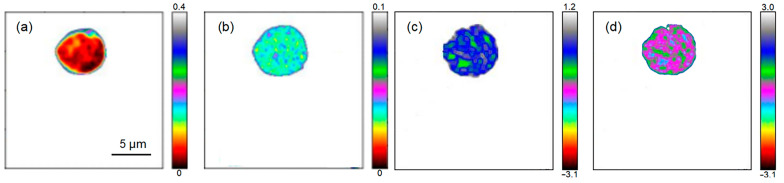
The results of processed THz s-SNOM image of individual Arabidopsis chloroplasts. (**a**,**b**) present the near-field amplitude images from 2 Ω to 5 Ω; (**c**,**d**) are the near-field phase images from 2 Ω to 5 Ω.

**Figure 7 ijms-24-13630-f007:**
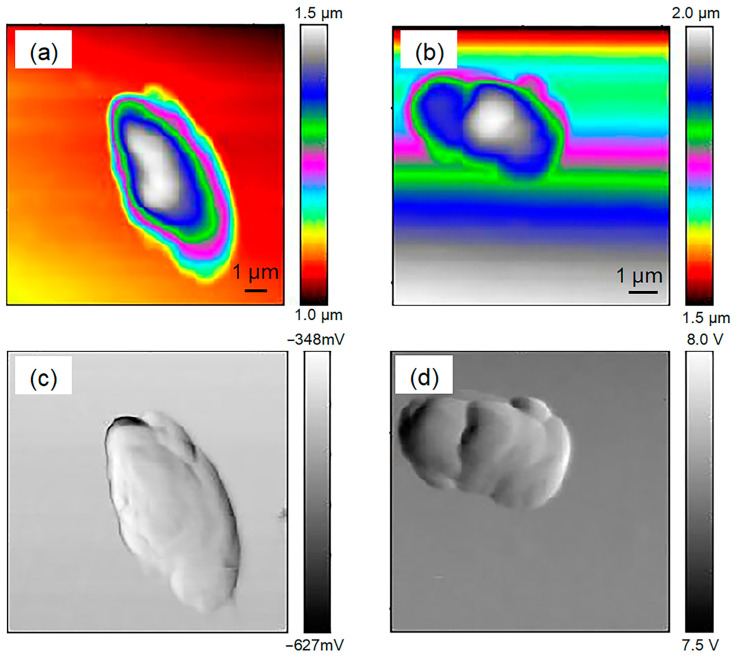
The AFM imaging of individual chloroplasts from two tea cultivars. (**a**,**b**) Topographical images of chloroplasts from ‘*Echa 10*’ and ‘*Shuchazao*’. (**c**,**d**) Amplitude images of chloroplasts from ‘*Echa 10*’ and ‘*Shuchazao*’.

**Figure 8 ijms-24-13630-f008:**
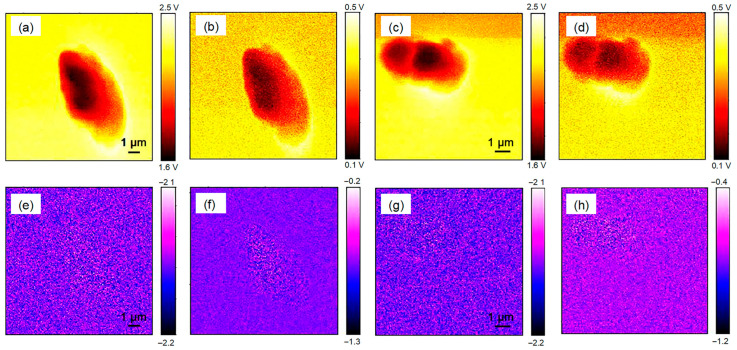
The THz s-SNOM image of individual chloroplasts from two tea cultivars. (**a**–**d**) present the near-field amplitude chloroplast images of ‘*Echa 10*’ (**a**,**b**) and ‘*Shuchazao*’ (**c**,**d**) from 2 Ω to 5 Ω; (**e**–**h**) are the near-field phase chloroplast images of ‘*Echa 10*’ (**e**,**f**) and ‘*Shuchazao*’ (**g**,**h**) from 2 Ω to 5 Ω.

**Figure 9 ijms-24-13630-f009:**
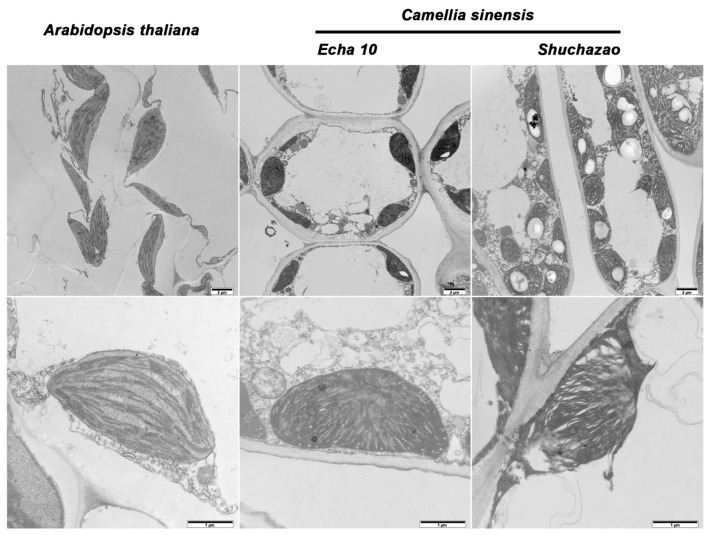
The ultrastructure images of chloroplasts from *Arabidopsis thaliana* and *Camellia sinensis* using TEM.

**Figure 10 ijms-24-13630-f010:**
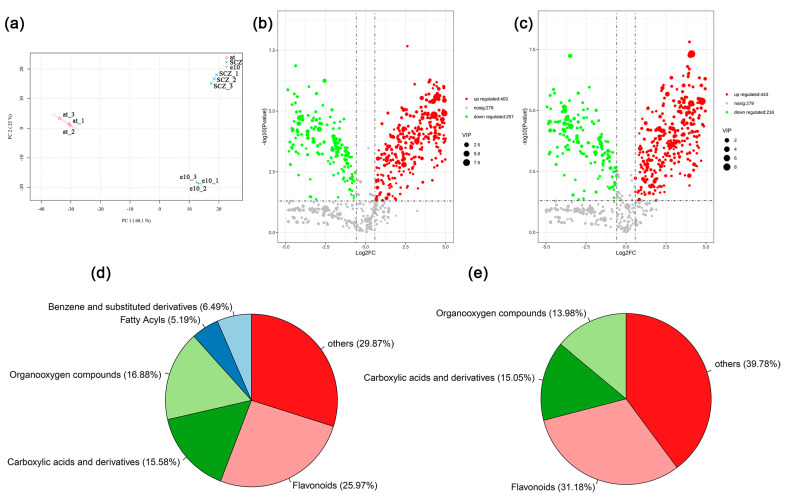
The metabolomic analyses of leaves from *Arabidopsis thaliana* and *Camellia sinensis*. (**a**) PCA analysis. (**b**,**c**) Identification of differential metabolites in ‘*Echa 10*’ vs. *Arabidopsis thaliana* (**b**) and ‘*Shuchazao*’ vs. *Arabidopsis thaliana* (**c**). (**d**,**e**) Classification of differential metabolites in ‘*Echa 10*’ vs. *Arabidopsis thaliana* (**d**) and ‘*Shuchazao*’ vs. *Arabidopsis thaliana* (**e**).

**Figure 11 ijms-24-13630-f011:**
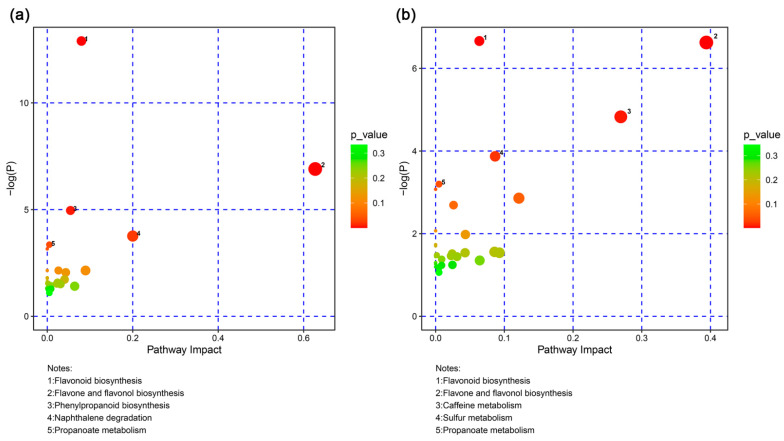
The MetPA analyses of differential metabolites in ‘*Echa 10*’ vs. *Arabidopsis thaliana* (**a**) and ‘*Shuchazao*’ vs. *Arabidopsis thaliana* (**b**).

## Data Availability

Not applicable.

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
