# Peer review of "Organelle Imaging with Terahertz Scattering-Type Scanning Near-Field Microscope"

_ijms, 2023, doi:10.3390/ijms241713630_

Round 1
Reviewer 1 Report
Terahertz radiation has many advantages for imaging the biologic objects including low quanta energy and large penetration depth. Still, traditional optical microscopic technique extrapolated to THz frequency range result in very low lateral resolution limited by the wavelength. Near field microscopy is one of the ways to surpass the diffraction limit in THz imaging. The paper under consideration describes application of this approach to particular small sized biological objects. The manuscripts provides detailed description and analysis of the related challenges and image processing techniques that facilitate overcoming these challenges. Minor English language polishing should be done in order to make the presentation a bit more clear, but otherwise the paper is ready and worth immediate publication.
Minor English language polishing should be done in order to make the presentation a bit more clear
Author Response
Thanks for your review. The manuscript has been carefully revised word by word.
Reviewer 2 Report
The manuscript describes the application of an advanced THz scattering near-field imaging method for detecting chloroplasts, on gold substrate with nano-flatness, combining with an image processing method to remove the background noise towards obtaining the subcellular grade internal reticular structure in the Arabidopsis chloroplast THz image.
The topic is novel, the scope is adequate and the methods are appropriate. The results represent significant improvement in THz imaging technique in subcellular organelle imaging.
The manuscript is well prepared with logical introduction introducing the background, problem statement and objectives. Over all adequate English usage.
A minor revision is recommended to correct minor issues.
Some suggestions:
1. a space is needed before unit, for example 3 mm.
2. Please explain “Rayleigh diffraction criterion”. And how does the current technique reported in the work overcome this, or not?
3. Line65-68 needs references.
4. Line 102, obtained -> obtain
5. Section 2.4, 2.7 lack the applied operating parameters.
6. L216, was sample dehydrated in a graded series of ethanol solutions altogether for 15 min or in each solution for 15 min?
7. Fig. 1, what are the short red line and square beneath reference mirror representing?
8. Some images lack readable scale bars and some do not have scale bars.
9. Fig. 4-6 caption does not state which sample.
10. Fig. 4, a-d are at 2Ω to 5Ω, so a at 2 Ω, d at 5 Ω. And b and c?
11. Tables S1 and S2 are mentioned but they are not provided.
12. Line 157, 1.550, use . as decimal point.
13. The highly impacting work applying THz spectroscopy to study biomedical materials as well as background of Terahertz radiation introduced herein, should be cited doi:10.1038/ncomms9631
Author Response
The manuscript describes the application of an advanced THz scattering near-field imaging method for detecting chloroplasts, on gold substrate with nano-flatness, combining with an image processing method to remove the background noise towards obtaining the subcellular grade internal reticular structure in the Arabidopsis chloroplast THz image.
The topic is novel, the scope is adequate and the methods are appropriate. The results represent significant improvement in THz imaging technique in subcellular organelle imaging.
The manuscript is well prepared with logical introduction introducing the background, problem statement and objectives. Over all adequate English usage.
A minor revision is recommended to correct minor issues.
Response: Thanks for your review. We have provide point-by-point responses below.
Some suggestions:
- a space is needed before unit, for example 3 mm.
Response: We have check the manuscript carefully and revised the error.
- Please explain “Rayleigh diffraction criterion”. And how does the current technique reported in the work overcome this, or not?
Response: When the center of the diffraction pattern of a certain object point coincides with the first minimum of the diffraction pattern of another object point, the light intensity between the centers of the two diffraction patterns is about 80% of the center maximum. At this point, the distance between the two object points is regarded as the minimum distance that optical instruments can distinguish. The simple formula is 1.22 λ/D. Considering the wavelength of THz waves (0.1 - 10 THz) is about 3 mm - 30 μm, the resolution of tradition far-field imaging is insufficient for cells and molecules.
The scheme based on THz-s-SNOM belongs to the classical tip scattering method. In 1928, Synge et al. proposed an optical super-resolution theory of using a needle tip to realize near-field scanning imaging (Synge et al., 1928, 6(35): p. 356-362; Philosophical Magazine). The enhancement effect of a metallized probe tip in THz-SNOM primarily relies on the lightning rod effect. The lightning rod effect is not limited by the spectral frequency and can enhance the local electric field in a wide frequency range. The spatial resolution of the system depends on the radius of curvature of the probe tip rather than the wavelength of incident electromagnetic wave. The resolution can reach to nanometer scale.
- Line65-68 needs references.
Response: Thanks for reminding. We have added references in the manuscript. Since the journal editor adjust the format according to the journal style, We can hardly insert references without changing the format. Thus, we added the detail information of references at the corresponding positions in the manuscript.
- Line 102, obtained -> obtain
Response: Thanks for your kind reminding. We have revised the mistake.
- Section 2.4, 2.7 lack the applied operating parameters.
Response: The applied operating parameters in fluorescent microscope and TEM imagings have been added in the sections 2.4 and 2.7.
- L216, was sample dehydrated in a graded series of ethanol solutions altogether for 15 min or in each solution for 15 min?
Response: Thanks for your reminding. The mistake have been revised.
- Fig. 1, what are the short red line and square beneath reference mirror representing?
Response: The short red line and square beneath is caused by mistakes, which have been corrected in the Fig. 1.
- Some images lack readable scale bars and some do not have scale bars.
Response: The bars in all figures have been carefully reset. Some images share the bars with adjacent images, thus appear no bars.
- Fig. 4-6 caption does not state which sample.
Response: Thanks for reminding. The sample information have been added in the Fig. 4-6 captions.
- Fig. 4, a-d are at 2Ω to 5Ω, so a at 2 Ω, d at 5 Ω. And b and c?
Response: The statement have been revised.
- Tables S1 and S2 are mentioned but they are not provided.
Response: We previously uploaded supplementary files including tables S1-S5. We upload them again this time.
- Line 157, 1.550, use . as decimal point.
Response: Thanks for reminding. The statement has been revised in a proper way.
- The highly impacting work applying THz spectroscopy to study biomedical materials as well as background of Terahertz radiation introduced herein, should be cited doi:10.1038/ncomms9631
Response: Thanks for reminding. The valuable work has been added in the manuscript.

Reviewer 3 Report
The statement should be clarified or removed from the abstract:
The preliminary results suggested that the interspecific different THz information is probably related to metabolite 27 differences among species.
If »probably« is not confirmed than the manuscript is not suitable for the publication.
Fig 10. and 11: how big is the experimental error?
Author Response
The statement should be clarified or removed from the abstract:
The preliminary results suggested that the interspecific different THz information is probably related to metabolite 27 differences among species.
If »probably« is not confirmed than the manuscript is not suitable for the publication.
Response: Thanks for your suggestions. According to the anatomical and metabolic results, the different THz information is related to metabolite differences among species. The statement have been revised and many studies with the similar results have been presented in the 'discussion' section.
Fig 10. and 11: how big is the experimental error?
Response: The original quantitative data of metabolites have been uploaded in supplementary files and PCA analyses have been displayed in figure 10A. The repeatability of biological repeats meets the requirement.

Round 2
Reviewer 3 Report
can be published